# Feasibility and Reliability of a Physical Fitness Test Battery in Individuals with Down Syndrome

**DOI:** 10.3390/ijerph16152685

**Published:** 2019-07-27

**Authors:** Ruth Cabeza-Ruiz, Francisco Javier Alcántara-Cordero, Isaac Ruiz-Gavilán, Antonio Manuel Sánchez-López

**Affiliations:** 1Department of Human Motricity and Sports Performance, University of Seville, 41013 Seville, Spain; 2Research Support Staff, Faculty of Education, University of Seville, 41013 Seville, Spain; 3Research Fellow, Department of Human Motricity and Sports Performance, University of Seville, 41013 Seville, Spain

**Keywords:** Down syndrome, psychometry, physical fitness, reliability, feasibility

## Abstract

Background: Down syndrome (DS) is a genetic disorder that occurs because of an abnormal division between cells that results in an extra copy of chromosome 21. Some studies show that physical exercise in people with DS increases some cognitive capacities, such as memory, and improves the quality of life. Aim: The main aim of this study was to perform an analysis of the reliability and feasibility of the SAMU-Disability Fitness (DISFIT) battery in adults with DS. Methods: A cross-sectional study with a test–retest design was performed in a maximum interval of 2 weeks in 37 subjects (11 women and 26 men) aged between 21 and 58 years old with DS. Eight field-based fitness tests were proposed to assess the physical fitness (PF) of adults with DS: Body Mass Index (BMI), Waist Circumference (WC), the Timed Up and Go test (TUG), the Deep Trunk Flexibility test (DTF), the Hand Grip test (HG), the Timed Stand Test (TST), the 30-s Sit-Up (SUP) and the 6-Min Walk Test (6MWT). Results: The intra-class correlation coefficient (ICC) in all the tests was good and high (>0.80), except for the 6MWT, whose reliability was fair. Conclusion: The SAMU-DISFIT battery is a reliable and feasible physical fitness battery which has been created with the purpose of establishing tests which measure the four basic components of PF (flexibility, cardiorespiratory fitness, musculoskeletal fitness and motor fitness) in adults with DS.

## 1. Introduction

Down syndrome (DS) is a genetic disorder that occurs because of an abnormal division between cells that results in an extra full or partial copy of chromosome 21. In most cases, the extra copy is maternally derived through an error in cell division named non-disjunction. This error can be originated during meiosis I, during meiosis II, or from a mitotic error. Other causes are when only a segment of chromosome 21 has three copies (partial trisomy) or when the whole chromosome is triplicated but only a proportion of the cells are trisomic (mosaicism), with other cells being normal [1]. The characteristics of this population include a high rate of heart problems [2], a high body mass index, numerous musculoskeletal deficits which limit their capacity to perform daily activities [3] and an increased risk (80% at 65 years old) of developing a dementia of Alzheimer’s type [4].

Moreover, people with DS seem to have a lower physical fitness (PF) than their peers without disabilities [5] and also than their peers with intellectual disabilities (ID) without DS [6,7]. From the health point of view, their low PF is one of the most important problems to solve in this population [8]. Their low level of strength and resistance causes a high energy use during daily activities, which brings about tiredness and a sedentary lifestyle. Physical inactivity in turn frequently entails overweight and obesity among the majority of this population [9].

It has been demonstrated in various studies that physical exercise, regardless of its intensity, decreases the rate and the seriousness of diseases derived from physical inactivity (such as cardiovascular diseases, hypertension, dyslipidemia or type II diabetes, among others) and fosters organic adaptations which positively impact on the autonomy of people with DS [9,10,11].

The practise of physical activity would therefore improve the quality of life of people with DS. Furthermore, some studies show that physical exercise also increases some cognitive capacities, such as memory [12]. Lastly, as they are dependent people, the improvement of their autonomy positively impacts on the quality of life of the people around them [13].

Following this line, and although much research has studied PF in subjects with DS, there are not many studies where the feasibility and reliability of the tests used for its assessment have been evaluated. In our study, different physical fitness tests were selected in order to analyse psychometric properties in adults with DS. With them, the PF level of people with DS can be determined in order to be able to establish programs in accordance with their needs. These help to prevent the negative effects of a bad physical condition and not very healthy lifestyles.

The main goal of this study is to conduct an analysis of the reliability and feasibility of a battery of tests of the assessment of PF in adults with DS.

## 2. Methods and Materials

### 2.1. Study Design

A cross-sectional study with a test–retest design was performed in a maximum interval of 2 weeks. People with DS were recruited from 12 care centres for people with ID in Seville (Spain).

### 2.2. Participants

A total of 37 subjects with DS (11 women and 26 men) aged between 21 and 58 years old with mild/moderate ID were evaluated. Individuals were included if they fulfilled the following inclusion criteria: (i) diagnosed with Down syndrome by the official competent administration; (ii) institutionalised in day care centres; (iii) able to follow simple verbal instructions; (iv) can walk autonomously; and, (v) have medical authorisation which reflects their aptitude to carry out physical activity without risks to their health. The legal tutors of the individuals interested in participating in the study signed a participation consent form. The study was approved by the Ethical committee of Biomedical Research of Andalusia (Seville, Spain) and follows the Helsinki guidelines for ethical behaviour [14].

### 2.3. Procedures

The tests were performed in a safe and familiar environment (in their care centres) by the same research team (10:00–13:00), always with the attention of one or various caregivers at the centre. The participants performed all the tests in 2 non-consecutive days with a maximum interval of 2 weeks between them. Based on the American College of Sports Medicine (ACSM) guidelines, the evaluators gave simple and clear verbal instructions in order to facilitate the participants’ comprehension. Furthermore, they made prior demonstrations if the subject needed them. A period of familiarisation was not allowed prior to the tests to avoid the learning effect on the final results. During all the measurement processes, they were given positive and individualised feedback. The purpose was to guarantee the maximum performance of the participants and avoid the usual distraction of this group; standardised protocols and instructions were eschewed [15]. The measurements were not continued if a subject refused to participate during any part of the study. Any type of execution error was noted for the feasibility test computation.

### 2.4. Measurements

The tests were selected based on their psychometric properties, their low cost and having been previously used both in subjects with DS [10,16,17], ID [18,19] and in other populations [20,21]. Eight field-based fitness tests were proposed to assess the PF of adults with DS. Firstly, the characteristics with respect to body composition were evaluated (2): Body Mass Index (BMI) and Waist Circumference (WC). After that, motor and musculoskeletal fitness were tested in this order: The Timed Up and Go test (TUG), the Deep Trunk Flexibility test (DTF), the Hand Grip test (HG), the Timed Stand Test (TST), and the 30-s Sit-Up (SUP). Finally, the 6-Min Walk Test (6MWT) was used to measure the cardiorespiratory fitness. The complete battery was called the SAMU Disability Fitness Battery (SAMU-DISFIT).

### 2.5. Body Composition (2)

The BMI was determined from the weight and the height of the subjects (i.e., kg/m2). Height was measured with a standard stadiometer and weight was tested with a scale. The instructions proposed by Suni et al. [21] were followed for the measurement of size and WC. The measurement was taken at the medium point between the edge of the 10th rib and the iliac crest to determine the WC of the participants. Two measurements were performed and the average between them was taken. If the difference was greater than a centimetre, a third trial was performed.

### 2.6. Motor Fitness (1)

The TUG was measured to determine agility, dynamic balance and capacity of autonomous movement. When signalled to do so, the subject got up and moved as soon as possible to a line situated at a distance of 3 m and returned to the starting position, covering a total of 6 m [22]. The fastest result out of two was noted. A trial run was allowed prior to carrying out the test. A modification of this test (comfortable speed) has shown good reliability results in people with intellectual disability (ID) [23]. The results of these tests were related with the risk of falls.

### 2.7. Musculoskeletal Fitness (4)

The DTF was used to value multiarticular flexibility. The initial position of the subjects was standing with their legs separated at the width of the waist. The subjects flexed their trunk and their knees and put their arms downwards and behind their legs to slide the cursor of the millimetre bar with their fingertips of both hands. A measurement bench (10 × 10 × 55 cm, height, width and length, respectively) was used for the evaluation. Two tries were done, and the best mark was chosen, following the indications of Valdivia et al. with children and adolescents [24].

The HG was used to measure the strength of the hand and of the forearm muscles, especially of the finger flexors [20,25,26]. The participants were asked to push as hard as possible after the command ‘and go’. There were two tries with each hand, establishing 10 s of rest between them [27].

The TST is a reproducible measure of lower extremity functionality. The participants completed 10 squats as quickly as possible with their arms crossed at the height of the chest. The best of 2 tries was selected for statistical analysis. A modification of this test has been used before in people with ID [28]. It serves to quantify the resistance strength of the musculature of the lower limbs (extension of hip and knee).

The SUP estimates the resistance strength of the muscles of the abdomen and of the hip flexors. The aim was to perform the maximum number of repetitions in 30 s [29]. As Boer and Moss did for people with DS, the participants laid down on a mat in the supine position with their knees flexed, their feet pressed on the ground and their hands on their thighs. To complete the tests, the participants had to slide their hands along their thighs to their kneecaps and then return to the initial positions [16].

### 2.8. Cardiorespiratory Fitness (1)

The 6MWT is an automatic rhythm test which requires a person to walk as far as possible in 6 min on a slope-free floor and without running [30]. The cardiorespiratory aptitude is evaluated with devices with GPS technology (Polar M200, Kempele, Finland). The distance covered during the test, measured in metres, was used as a result measure. Only one try was allowed.

### 2.9. Statistical Analysis

The participants’ data were analysed to check the distribution normality and the homoscedasticity (Levene’s test). A mixed factorial ANOVA was performed with test–retest measurements such as the intra-subject factor and the sex as the variable between the groups to find possible differences between the groups (men and women) and between the conditions (test–retest).

The feasibility ending rates were calculated according to what was described by Wouters et al. [15], i.e., <50% not feasible, 50–75% quite feasible and >75% feasible. The reliability of the SAMU-DIS FIT test–retest was determined by the intra-class correlation coefficient (ICC) (mixed two-way model, absolute agreement) with 95% confidence intervals. The ICC was interpreted as follows: values of 0.90–0.99 reflect high reliability, 0.80–0.89 good reliability, 0.70–0.79 fair reliability and points equal to or lower than 0.69 scant reliability [31].

The indications of Atkinson et al. [32] were followed to calculate the standard error of measurement (SEM). This means an absolute reliability and the degree to which the repeated measurements varied in the subjects between the test and the retest. The Minimal Detectable Change (MDC) was also estimated. This shows the minimum change necessary between evaluations that is not a consequence of a measurement error [15].

## 3. Results

The results of the sample recruitment can be observed in the flow diagram (Figure 1). In a first phase, there were 753 adults with intellectual disabilities, of whom 715 were excluded for the following reasons: they did not fulfil the inclusion criteria (not diagnosed with Down syndrome), they rejected taking part in the study or were excluded for other reasons related with limitations in the receiving/comprehension of the information, illnesses or feeling ill. Thirty-eight people with DS (26 men and 12 women) were selected from this sample for the study. During the follow-up, there was one loss (a women), so the dropout rate was 2.63% in relation with the final participation in each of the tests. There were no losses in the data analysis so, finally, the sample consisted of 37 people with trisomy 21 (mean: 37.57 years old, height 149.75 m, weight 68.47 kgs).

The result of the mixed factorial ANOVA revealed that there were no significant differences between the average of each of the variables, nor between groups (men/women), nor between conditions (test/retest) (*p* > 0.05). Table 1 shows the results and the ICC of each of the variables in the test and the retest. In all the tests, the ICC was good and high (>0.80), except for the 6MWT, for which the reliability was fair.

Leaving aside the tests of body composition, the tests with the best results of ICC were the HG (ICC = 0.90), the TUG (ICC = 0.89) and the DTF (ICC = 0.87). Moreover, all the SEM values achieved the criterion (SEM < SD 1/2), proposing an admissible measurement precision and all the feasibility results were excellent (>97%), except for the SUP (82.45%).

## 4. Discussion

In this work, the findings show a high reliability and feasibility of the tests proposed, so the SAMU-DISFIT battery is shown to be a recommendable tool to evaluate adults with DS. Safety, easiness and simplicity were other objectives which have been achieved.

In the scientific literature, there exists research with the aim to determine the PF level of people with DS. According to the systematic review carried out by Ayán et al. [10], most of them have been centred on performing a series of tests and describing the PF data obtained without previously checking the reliability and feasibility of the tools used in this population. In other cases, the test–retest has been analysed in one or various isolated tests—the majority obtained from a specific physical capacity. For example, the 6MWT [17,33,34] and the HG [35,36] are two tests that the reliability of which has been evaluated previously in adults with DS; however, they have not been supplemented with feasibility data.

The reliability of the TUG, SUP and TST has been checked before. Villamonte et al. [37] performed the TUG test under the keyword of moving at a “comfortable speed”. Their results showed low ICC values (0.22–0.24). In our study, we performed the TUG with the subject moving as quickly as possible as our experience is that people with intellectual deficits understand specific information (the quickest possible) better than tests with subjective information (comfortable speed). These adaptations offered more reliable data (ICC = 0.89, feasibility 100%).

As for the SUP and TST, both have been used previously as part of various batteries of tests but also with other protocols. In the work of Boer and Moss [16], the reliability values of the SUP were very high (0.98) despite the participants performing as many curl-ups as possible for as long as possible. Regarding previous reliability results of the TST, Villamonte et al. [37] conducted the test valuing the number of repetitions that the subject could perform in 20 s, but they obtained poor and fair ICC values (0.54 and 0.76, men and women, respectively). None of these works contributes feasibility data. In this study, the tests proposed show very good reliability data and incorporate feasibility data for the first time.

Regarding flexibility, most of the articles reviewed use the Sit and Reach, or adaptations of it, to measure this capacity in people with ID [38,39,40]. Our study is the first to use the DTF in people with DS, with excellent reliability (ICC = 0.89) and feasibility (97.3%) results. This test was chosen for this study for various reasons. The first is that it is a test with an execution technique that is simpler, as the participants do not have to keep their knees completely extended. Secondly, the motion to be performed is a more natural movement in daily life than the traditional Sit and Reach.

With respect to psychometric properties, it has been highly recommended to emphasise providing information on measurement error as it contributes additional information to the ICC [41]. In a bibliographic review on PF tests in people with DS [10], it was concluded that although there exists much research, the tests on feasibility and reliability are scarce and are mainly performed in groups of young people. In this work, we also present MDC values which supplement those obtained in the ICC, SEM and feasibility.

To the best of our knowledge, this is the first time that a battery of tests has been proposed for the evaluation of PF in adults with DS that shows reliability and feasibility data. Salaun and Berthouze-Aranda [42] carried out a study with people with ID, whose PF they evaluated with the Eurofit Battery. This battery contains numerous tests and, although the authors excluded the most demanding, those that they selected for their study did not contribute feasibility data but had an evident complexity of execution that could be an important barrier for people with ID, with or without DS.

On the other hand, although there exist batteries adapted to people with ID such as the FUNfitness or the Eurofit Special, there are few works which value their psychometric characteristics. With regard to the FUNfitness, there is currently only one study which evaluates the reliability and the feasibility of the tests. However, this study does not value all the tests included in the FUNFitness but only those of strength and balance [43]. As for the Eurofit Special battery, the reliability and feasibility were evaluated in a study performed with 1602 persons. Yet, we have not been able to consult the bibliographic source as it was inaccessible (perhaps due to an error in the references offered by the authors). Furthermore, one of the main limitations of the study, pointed out by the authors themselves, is that it does not distinguish participants with DS. Therefore, although these batteries could be appropriate to evaluate the PF of people with ID, it is necessary to check that their reliability and feasibility is good in more homogeneous samples, specifically in people with DS [29].

There exists a battery especially designed for elderly adults with ID (50–89 years old), which was performed by Hilgenkamp, Van Wijck and Evenhuis [35]. However, like the previous ones, the studies on this battery’s psychometric aspects do not differentiate between participants with trisomy 21 or without it.

Specifically, in children and adolescents with DS, Tejero et al. [36] valued the reliability of other batteries of tests. These tests had been designed for people without disabilities and although the reliability results were good, they do not contribute feasibility data and no tool of measurement of flexibility was assessed.

On the other hand, Villamonte et al. [37] measured the psychometric properties of 16 balance tests in children, teenagers and young adults with DS. Among the tests conducted are the STS (20 s long) and the TUG (9 m distance), the former being reliable (ICC > 50) only in young women and young men, while the latter is not reliable for any group of DS. Among the limitations of the study is its small sample (21 people) and a very varied age range (5–31 years old).

To date, only one study evaluates various components of the PF exclusively with adults with DS [25]. Although the authors used a battery of functional tests with 371 people with DS from South Africa, the feasibility and reliability results were published with a sample of 43 people [16]. The limitation of their proposal is that as it is a battery made up of numerous tests, a long time is needed to evaluate each person. Also, none of these tests shows feasibility results, so the percentage of people with DS who could perform the technique successfully is not known.

The main limitation of this study has been the difficulty in attracting subjects with DS due to the small population existing with this alteration who fulfils the inclusion criteria. In future research, it is necessary to recruit broader samples, especially of women, to be able to carry out statistical analyses for different age groups.

## 5. Conclusions

The SAMU-DISFIT battery is a reliable and feasible physical fitness battery that has been created with the purpose of establishing tests which measure the four basic components of PF (flexibility, cardiorespiratory fitness, musculoskeletal fitness and motor fitness) in adults with ID, with or without DS.

The SAMU-DISFIT battery will serve in future research as a tool to establish reference values of PF levels in people with DS and, thus, enable carrying out intervention programs of physical activity and health adapted to the characteristics of these people. These can be implemented in care centres with a view to improving their quality of life.

## Figures and Tables

**Figure 1 ijerph-16-02685-f001:**
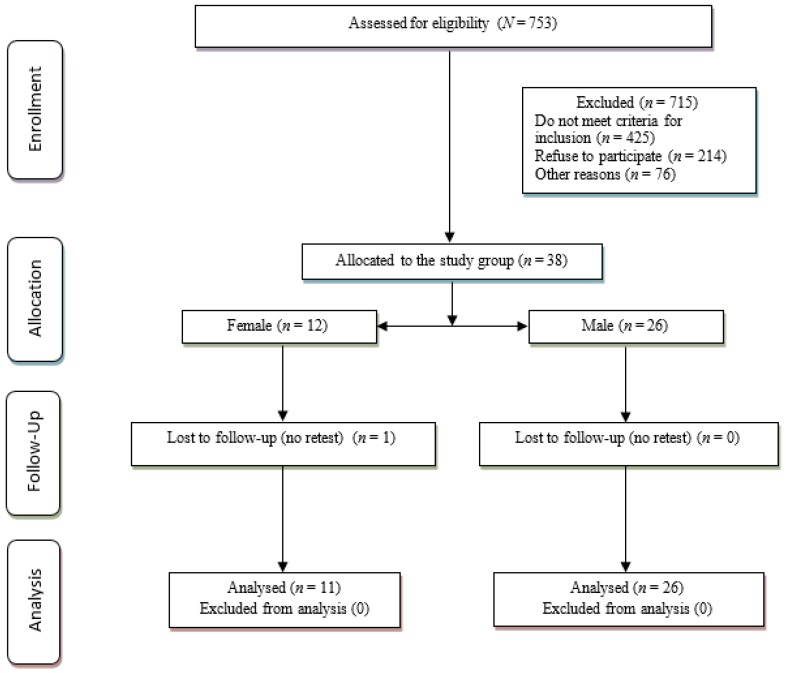
Flowchart of the results of the sample recruitment.

**Table 1 ijerph-16-02685-t001:** Test–retest reliability in adults with Down syndrome.

Physical Fitness Component	Test	*n*	TEST Mean (SD)	RE-TEST Mean (SD)	ICC (95% CI)	SEM	MDC	FEASIBILITY %
Body composition	Waist Circumference (cm)	37	94.94 (14.17)	95.51 (15.23)	0.96 (0.928–0.981)	2.83	7.84	98.65
	Body Mass Index (kg/m^2^)	37	30.75 (6.42)	30.58 (6.56)	0.99 (0.971–0.992)	0.79	2.20	100
Motor fitness	Timed Up and Go test (s)	37	5.07 (1.01)	4.92 (0.99)	0.89 (0.787–0.942)	0.33	0.93	100
Musculoskeletal fitness	Deep Trunk Flexibility test (cm)	37	32.06 (7.42)	32.80 (7.12)	0.87 (0.764–0.933)	2.60	7.21	97.3
	Hand Grip (kg)	37	21.41 (6.25)	21.54 (6.67)	0.90 (0.810–0.948)	2.05	5.69	100
	10 Timed-Stand Test (s)	37	21.94 (6.34)	20.63 (6.19)	0.80 (0.651–0.894)	2.77	7.69	97.3
	30-s Sit-Up (number)	37	15.22 (5.27)	15.48 (5.11)	0.80 (0.628–0.899)	2.31	6.40	82.45
Cardiorespiratory fitness	6-Min Walk Test (m)	37	463.08 (82.14)	457.44 (93.73)	0.77 (0.595–0.874)	42.26	117.14	98.65

Notes: SD = standard deviation; ICC = intraclass correlation coefficient; CI = interval confidence; SEM = standard error of measurement; MDC = minimal detectable change.

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
