# Peer review of "Feasibility and Reliability of a Physical Fitness Test Battery in Individuals with Down Syndrome"

_ijerph, 2019, doi:10.3390/ijerph16152685_

Reviewer 1 Report

This manuscript looks interesting, since it is important to improve the life quality and fitness of patients, diagnosed with Down Syndrome. It would be nice in the future, if you could extend this study by recruiting more patients.

Some suggestions:

In the introduction you may mention more on genetic factors, involved in Down Syndrome.

DS may be risk factor for neurodegenerative diseases, such as Alzheimer’s disease too by the trisomy of chromosome 21. You may mention it in the introduction.

In materials and methods, in participants chapter you may add a table on demographics of patients.

How about the cognitive functions of patients? It may be nice to discuss that these tests may prevent the cognitive dysfunctions in DS patients.

Author Response

Dear reviewer,

Thank you for the interesting contributions that make our work considerably better.

In the following lines we try to solve/respond to your considerations towards our study.

Point 1: It would be nice in the future, if you could extend this study by recruiting more patients.

Response 1: One of our main goals for the future is to  considerably increase the sample size.

Point 2: In the introduction you may mention more on genetic factors, involved in Down Syndrome.

Response 2: Included from lin 2 to 7

Point 3: DS may be risk factor for neurodegenerative diseases, such as Alzheimer’s disease too by the trisomy of chromosome 21. You may mention it in the introduction.

Response 3: Included

Point 4: In materials and methods, in participants chapter you may add a table on demographics of patients.

Response 4: In order to not include tables that do not provide too much information or that this is not relevant in the article, the authors decided not to insert a table with the characteristics of the sample in relation to age, weight and height, since these data can be observed in the description of the participants (line 59, 155, 156) and in the BMI variable (table 1). If the reviewer considers it fundamental, we will include it.

Point 5: How about the cognitive functions of patients? It may be nice to discuss that these tests may prevent the cognitive dysfunctions in DS patients.

Response 5: The use of the tests does not influence the cognitive capacity of the participants. The tests only indicate the physical condition of the participants. Unfortunately, in our study we could not measure the cognitive abilities of the participants (we only know the level of ID: mild/moderate) so we can not make correlations between cognitive capacity and physical fitness based on the results of the tests.

Reviewer 2 Report

In this paper named" Feasibility and reliability of a physical fitness test battery in individuals with Down syndrome", the authors addressed very relevant point related to Down Syndrome (DS) where they tried to do the study on the reliability and feasibility of a battery of Physical Fitness (PF) assessment tests in adults with DS to prove that physical exercise in people with DS increases some capacities like memory, and improves the quality of life. They showed very interesting observations in a very well manner. My recommendation for this manuscript is to get published in the present form. 

Author Response

Dear reviewer,

We would also like to thank you for your kind words towards our study.

Reviewer 3 Report

This is a reasonable paper aimed at validating certain tests of performance in Spanish subjects with Down syndrome. As the authors suggest, a major limitation is the small sample size, especially for females. More information on the large number of exclusions would be helpful. Describe why so many did not meet inclusion criteria. Make paper consistent with past tense; statistical section, for example is present tense. Otherwise, I think the paper is acceptable with many English/spelling corrections/suggestions noted below.

Author list line 6: Is the email address correct?

 Line 15: Omit second PF and introduce SAMU-DISFIT here. Should physical fitness be capitalized?

Check names and capitalization of the tests. For example, is the 6 minutes walk test not the 6 minute walk test?

Line 33: PF should be physical fitness.

Line 50: Change found out to determined.

Line 51: Programmes >programs

Line 58: Centres>centers and throughout rest of paper

Line 79: Clarify "eschewed"

Line 93: Reckoned up to determined

 Line 129: Explain devices with GPS technology.

Line 138 DISFIT not DIS-FIT

Line 178: Centered not centred

.Line 194: 20 seconds not 20s

Line 222: persons or subjects, not personas

Table: Check spelling of musculoskeletal.

References; Check all for consistency.  For example refs 6, 7, 19, 32, 34 and 40 are incomplete it appears.

Author Response

Reviewer 3

Dear reviewer,

Thank you for the interesting contributions that make our work considerably better.

In the following lines we try to solve/respond to your considerations towards our study.

Point 1: This is a reasonable paper aimed at validating certain tests of performance in Spanish subjects with Down syndrome. As the authors suggest, a major limitation is the small sample size, especially for females. 

One of our main goals for the future is to  considerably increase the sample size.

 Point 2: More information on the large number of exclusions would be helpful. Describe why so many did not meet inclusion criteria. 

Clarified from line 150  to line 152.

Point 3: Make paper consistent with past tense; statistical section, for example is present tense. 

The whole manuscript has been revised

Point 4: Otherwise, I think the paper is acceptable with many English/spelling corrections/suggestions noted below.

Some of the considerations related to the use of English language are due to the fact that we have used British English instead of American English since we  worked with a professional native british translator. In this sense we have changed every word that the reviewer suggested. 

Point 5: Author list line 6: Is the email address correct?

Author list and the corresponding affiliations reviewed

Point 6: Line 15: Omit second PF and introduce SAMU-DISFIT here. Should physical fitness be capitalized?

Corrected

Point 7: Check names and capitalization of the tests. For example, is the 6 minutes walk test not the 6 minute walk test?

Done

Point 8: Line 33: PF should be physical fitness.

Revised

Point 9: Line 50: Change found out to determined.

Changed

Point 10: Line 51: Programmes >programs

Done

Point 11: Line 58: Centres>centers and throughout rest of paper

Done

Point 12: Line 79: Clarify "eschewed"

The authors are sorry for not understanding what the reviewer 3 means by the following comment: Line 79: Clarify "eschewed".

Point 13: Line 93: Reckoned up to determined

Changed

Point 14:  Line 129: Explain devices with GPS technology.

Done

Point 15: Line 138 DISFIT not DIS-FIT

Changed

Point 16: Line 178: Centered not centred

Revised and changed in the entire document

Point 17: Line 194: 20 seconds not 20s

Done

Point 18: Line 222: persons or subjects, not personas

Revised

Point 19: Table: Check spelling of musculoskeletal.

Done

Point 20: References; Check all for consistency.  For example refs 6, 7, 19, 32, 34 and 40 are incomplete it appears.

Revised and corrected